# A New Staining Method Using Methionyl-tRNA Synthetase 1 Antibody for Endoscopic Ultrasound-Guided Fine-Needle Aspiration Cytology of Pancreatic Cancer

**DOI:** 10.3390/diagnostics15141783

**Published:** 2025-07-15

**Authors:** Sung Ill Jang, See Young Lee, Ji Hae Nahm, Jae Hee Cho, Jung Hyun Jo, Chan Min Jung, Beom Jin Lim, Jin Hong Lim, Hyung Sun Kim, Su Yun Lee, In Young Hong, Sunghoon Kim, Dong Ki Lee

**Affiliations:** 1Department of Internal Medicine, Gangnam Severance Hospital, Yonsei University College of Medicine, Seoul 06273, Republic of Korea; aerojsi@yuhs.ac (S.I.J.); seeyoung87@yuhs.ac (S.Y.L.); jhcho9328@yuhs.ac (J.H.C.); junghyunjo83@yuhs.ac (J.H.J.); maxed123@yuhs.ac (C.M.J.); lsy83@yuhs.ac (S.Y.L.); hdoll91@yuhs.ac (I.Y.H.); 2Department of Pathology, Gangnam Severance Hospital, Yonsei University College of Medicine, Seoul 06273, Republic of Korea; bjlim@yuhs.ac; 3Department of Surgery, Gangnam Severance Hospital, Yonsei University College of Medicine, Seoul 06273, Republic of Korea; doctorjin@yuhs.ac (J.H.L.); milk8508@yuhs.ac (H.S.K.); 4Institute for Artificial Intelligence and Biomedical Research, College of Pharmacy & College of Medicine, Gangnam Severance Hospital, Yonsei University, Seoul 06273, Republic of Korea; sungkim@biocon.snu.ac.kr; 5Yonsei Institute of Pharmaceutical Sciences, College of Pharmacy, Yonsei University, Incheon 21983, Republic of Korea; 6Department of Internal Medicine, Wonju Severance Christian Hospital, Yonsei University Wonju College of Medicine, Wonju-si 26426, Republic of Korea

**Keywords:** pancreatic neoplasm, pancreatic cancer, endoscopic ultrasound, cytology

## Abstract

**Background/Objectives**: Accurate determination of malignancy in pancreatic masses through endoscopic ultrasound-guided fine-needle aspiration (EUS-FNA) is crucial for appropriate clinical management and prognostic assessment. However, the diagnostic sensitivity of conventional cytology using Papanicolaou (Pap) staining remains limited, often leading to inconclusive results. In this study, we investigated the diagnostic utility of methionyl-tRNA synthetase 1 (MARS1) through immunohistochemical (IHC) and immunofluorescence (IF) staining as a potential biomarker for pancreatic cancer. IHC analysis was conducted on resected tissue samples from 10 patients, including both pancreatic ductal adenocarcinoma and corresponding non-neoplastic pancreatic tissue. Additionally, cytologic samples were obtained from 198 patients with pancreatic masses who underwent EUS-FNA for diagnostic evaluation. Pap staining and MARS1 IF staining were performed on liquid-based cytology slides derived from EUS-FNA specimens. **Results**: MARS1 was detected by IHC staining in the 10 surgical specimens diagnosed with pancreatic adenocarcinomas. After Pap staining, 37 patients were excluded because of unsuitable specimens, leaving 161 patients who underwent both Pap and MARS1 IF staining. EUS-FNA specimens from the 151 patients with pancreatic ductal adenocarcinoma were classified by Pap staining as atypia (n = 36), suspicious for malignancy (n = 55), or malignancy (n = 60). MARS1 IF staining was positive in 147 of these patients and negative in 4. MARS1 IF staining distinguished pancreatic cancer in specimens with atypia on Pap staining. The sensitivity for detecting pancreatic cancer was significantly higher for MARS1 IF staining than for conventional Pap staining (97.4% vs. 79.1%, *p* < 0.0001). **Conclusions**: The high sensitivity of MARS1 IF staining improved malignancy detection in pancreatic masses. Further prospective studies are required to validate our findings.

## 1. Introduction

Pancreatic ductal adenocarcinoma (PDAC) is the fourth leading cause of cancer-related deaths in the United States and has the worst prognosis of all major malignancies, with a 6% 5-year survival rate [1]. As most PDACs are unresectable at the time of diagnosis, an accurate pathologic diagnosis is necessary for guiding chemotherapy in unresectable PDAC. Currently, endoscopic ultrasound-guided fine-needle aspiration (EUS-FNA) is performed to overcome the drawbacks of percutaneous biopsy using abdominal ultrasonography or computed tomography and endoscopic retrograde cholangiopancreatography (ERCP). EUS-FNA has a sensitivity of 75–98%, specificity of 71–100%, positive predictive value (PPV) of 96–100%, negative predictive value (NPV) of 33–85%, and accuracy of 79–98% for diagnosing pancreatic cancer [2]. However, the rate of indeterminate cytology (i.e., “atypia” or “suspicious for malignancy”) in samples obtained by EUS-FNA is 8–17% [3,4].

Methionyl-tRNA synthetase 1 (MARS1) is a member of the aminoacyl-tRNA synthetase (ARS) family—enzymes essential for protein synthesis through the attachment of specific amino acids to their corresponding tRNAs [5,6]. Among ARSs, MARS1 plays a pivotal role in translation initiation by methionylating the initiator tRNA, a process required to begin mRNA translation [7]. Beyond its canonical function, MARS1 also responds to cellular stress; upon ultraviolet-induced DNA damage, it is phosphorylated at Ser662 and dissociates from ARS-interacting multifunctional protein-3 (AIMP3), thereby linking DNA damage signaling to global translational control.

MARS1 is highly expressed in multiple cancer types and is increasingly recognized for its role in tumorigenesis and cancer progression [7,8,9,10,11,12,13]. In pancreatic ductal adenocarcinoma (PDAC), elevated MARS1 expression correlates with poor prognosis, and in our previous study, it was identified—alongside lymph node metastasis—as an independent predictor of unfavorable outcomes in PDAC patients [14]. Mechanistically, MARS1 contributes to oncogenesis through various pathways: it enhances enzymatic activity in colon cancer cells [15], and overexpression of its substrate, initiator tRNA_i_^Met^, is sufficient to induce malignant transformation [16]. Furthermore, MARS1 stabilizes cyclin-dependent kinase 4 (CDK4) by forming a complex with heat shock protein 90, protecting CDK4 from proteasomal degradation and promoting cell cycle progression [5]. Conversely, suppression of MARS1 leads to CDK4 depletion and cell cycle arrest at the G0/G1 phase. MARS1 also competes with the tumor suppressor p16^INK4a for binding to the N-terminal domain of CDK4, further highlighting its potential oncogenic role [5].

Immunofluorescence (IF) staining using MARS1 antibody has demonstrated high sensitivity and accuracy in indeterminate biliary strictures [17]. The sensitivity of MARS1 IF staining was superior to that of conventional cytologic staining method (CCM) (i.e., Papanicolaou [Pap] stain) for endoscopically obtained brush cytology specimens (93.6% vs. 73.2%). Molecular assessment using MARS1 as cancer-specific marker significantly improved the diagnostic performance of brushing cytology because MARS1 IF staining differentiated the atypical cells observed in Pap staining [17]. In this study, we evaluated the diagnostic performance of MARS1 IF staining of pancreatic masses obtained by EUS-FNA.

## 2. Materials and Methods

### 2.1. Study Design and Participants

We retrospectively performed immunohistochemical (IHC) staining of pancreatic ductal adenocarcinoma and normal pancreatic tissues from patients who underwent surgery. We prospectively enrolled adults with a pancreatic mass who underwent EUS-FNA at our institution between October 2015 and April 2020. The inclusion criteria were age ≥ 19 years; pancreatic mass confirmed by computed tomography, magnetic resonance imaging, or positron emission tomography; and no prior procedure involving the papilla. The exclusion criteria were age ≤ 18 years, current pregnancy, intellectual disability, sensitivity to contrast agents, or previous pancreaticobiliary surgery. Final clinicopathologic diagnoses were based on pathologic diagnosis of surgical and/or EUS-FNA biopsy specimens or on clinical and radiologic data collected during ≥12 months of follow-up. Negative EUS-FNA cytology results were confirmed by data obtained during ≥12 months of follow-up.

The study protocol was approved by our Institutional Review Board (No. 3-2015-0234), and the study was registered at cris.nih.go.kr (accessed on 5 May 2020, KCT0004515). All participants provided written informed consent. All authors had access to the study data and approved the final manuscript.

### 2.2. MARS1 Immunohistochemical Staining in Surgical Specimens

MARS1 staining used immunohistochemical (IHC) staining and fluorescent staining. IHC staining was performed on surgical specimens, and cytology was performed using IF staining. MARS1 IHC staining was evaluated by light microscopy of whole slide fields. Positive MARS1 staining was defined as >10% staining of the cytoplasm of ductal epithelial cells, with moderate to strong intensity. Sections exhibiting dim or faint staining or <10% staining were considered MARS1 negative. Most sections exhibited clearly positive or negative immunoreactivity. Tissue processing and staining methods are described in the Appendix A.

### 2.3. EUS-FNA Cytology Samples

EUS-FNA was performed using a linear EUS scope (GF-UCT260; Olympus Medical Systems, Tokyo, Japan) and a 19- or 22-gauge needle (Wilson-Cook Medical, Inc., Winston-Salem, NC, USA). All EUS-FNA procedures were performed by 3 endoscopists with experience performing >400 EUS-FNA procedures. The needle was inserted into the pancreatic target lesion after visualization by EUS. After the stylet was removed, a 10 mL suction syringe was attached to the needle, and the needle was then moved forward and backward ≥ 15 times. The EUS–FNA was performed 3 times and tissue samples were individually collected to 3 sample tubes. The cytology samples were immediately transported to the cytology laboratory for liquid-based ThinPrep (Cytyc Corp, Marlborough, MA, USA) slide preparation, resulting in 6 slides from 3 sample tubes. Two slides were produced from one tube, for a total of six slides. One slide was collected from each tube, and Pap staining was performed on three slides, and MARS1 staining was performed on the remaining three slides. The latter were preserved in Pro-Fixx (Lerner Laboratories, Pittsburgh, PA, USA), wrapped in aluminum foil, and stored at 20 °C.

### 2.4. MARS1 Immunofluorescence Staining of Cytology Specimens

Pap staining was performed according to the manufacturer’s protocol. For MARS1 IF staining, the liquid-based ThinPrep slides were first permeabilized with 0.3% phosphate-buffered saline with Tween for 30 min. After washing 2–3 times with Tris-buffered saline with Tween (TBST), the slides were washed with distilled water and blocked with 3% goat serum at room temperature (RT) for 20 min. Anti-MARS1 primary antibody (0.2 mg/mL, 1:300; BICBIO INC., Suwon, Republic of Korea) was then added, and the slides were incubated at 37 °C for 1 h. The slides were again washed 2–3 times with TBST, after which the secondary antibody (Alexa Fluor 488; 1:300; Thermo Fisher Scientific, Eugene, OR, USA) was added, and the slides were incubated at RT for 30 min. After rewashing 2–3 times with TBST, 4′,6-diamidino-2-phenylindole (Thermo Fisher Scientific, MA, USA) was added.

### 2.5. Interpreting CCM (Pap) and MARS1 IF Stains

Cytologic samples were anonymized and grouped based on the staining method (CCM vs. MARS1 IF) for blinded evaluation. Diagnoses from slides stained with the conventional Pap method were categorized into six diagnostic groups: nondiagnostic, negative for malignancy, atypical, neoplastic (benign or other), suspicious for malignancy, and malignant [18]. Cases lacking adequate cellularity for interpretation were excluded, as both CCM and MARS1 IF staining require sufficient representative cells. For the purpose of analysis, the diagnostic outcomes were simplified into two groups: malignant (including malignant and suspicious for malignancy) and benign (including atypia and negative for malignancy).

Interpretation of MARS1 IF staining was guided by comparison to established controls—PANC-1 cells as positive (Appendix A) and CT26 cells as negative (Appendix A). Entire slide fields were examined under a fluorescence microscope (BX53; Olympus Corp., Tokyo, Japan) at magnifications of 200× or higher. A specimen was considered MARS1-positive if at least one cluster of cells displayed strong cytoplasmic fluorescence comparable to or exceeding that observed in the positive control. Specimens showing weak, diffuse, or ambiguous staining patterns were classified as negative. Two pathologists (JHN, BJL) independently evaluated all MARS1 IF–stained slides and were blinded to the clinical data and CCM results. Discrepancies were resolved by joint review of the specimen.

### 2.6. Study Outcomes

The primary endpoints were the diagnostic performance (i.e., sensitivity, specificity, PPV, NPV, and accuracy) of the MARS1 IF-staining method when used for pancreatic EUS-FNA specimens and comparison of the MARS1 IF and CCM staining results.

### 2.7. Statistical Analysis

Sensitivity, specificity, PPV, NPV, and overall accuracy were calculated using conventional statistical approaches. Comparisons of diagnostic performance between staining techniques were conducted using the Cochran–Mantel–Haenszel test and logistic regression with generalized estimating equations, an approach appropriate for correlated binary data. Statistical significance was defined as a two-sided *p* value of less than 0.05. All analyses were conducted using SPSS software (version 20; IBM Corp., Armonk, NY, USA).

## 3. Results

### 3.1. Characteristics of Patients

Formalin-fixed paraffin-embedded surgical samples from 10 patients, comprising 10 pancreatic ductal adenocarcinoma (PDAC) tissues and 10 matched normal pancreatic tissues, were obtained through the Yonsei Biobank at Gangnam Severance Hospital. All tumor specimens were histologically confirmed as PDAC. Immunohistochemical analysis revealed a significantly stronger MARS1 staining intensity in malignant tissues compared to normal pancreatic tissues (Appendix B Table A1, Appendix C Figure A1).

Additionally, a total of 198 patients presenting with a pancreatic mass were prospectively enrolled and underwent endoscopic EUS-FNA with subsequent cell block CCM staining (Figure 1).

A total of 4 patients were excluded because of nonpancreatic disease (common bile duct cancer, n = 2; colon cancer, n = 1; diffuse large B-cell lymphoma, n = 1), and 33 were excluded because of nondiagnostic specimens on CCM staining (insufficient or necrotic cytology). MARS1 IF staining was performed in 161 patients, all of whom were included in the final analysis.

The mean age of the 161 included patients was 65.6 years with a male/female ratio of 82:79 (Table 1).

The final clinicopathologic diagnosis was PDAC in 151 patients (93.8%), pancreatic cysts in 4 patients, and other pancreatic disorders in 6 patients. These diagnoses were established using EUS-FNA cytology and/or biopsy (n = 96), surgical specimens (n = 47), or clinical or radiologic criteria during ≥12 months of follow-up (median, 16.2 months; range, 12.5–25.6 months) (n = 7), their interpretation, as well as the experimental conclusions that can be drawn.

### 3.2. CCM Versus MARS1 IF Staining

On MARS1 IF staining, pancreatic cancers were defined by the presence of a strong green signal in the cytosol of cells (Appendix A), and benign specimens were defined by the presence of a dim green signal or no signal (Appendix A). We compared CCM with MARS1 IF staining of EUS-FNA specimens from 161 patients.

The distribution of diagnoses according to CCM were malignant, n = 63; suspicious for malignancy, n = 58; atypia, n = 39; and negative for malignancy, n = 1 (Table 2).

Among the 63 patients with specimens designated as malignancy, the final clinicopathologic diagnoses were PDAC (n = 60), neuroendocrine tumor (NET; n = 2), and acinar cell carcinoma (n = 1). Of the 60 patients with PDAC, MARS1 staining was positive in 58 and negative in 2 (Figure 2).

Among the 58 patients with specimens designated as suspicious for malignancy by CCM, the final clinicopathologic diagnoses were PDAC (n = 55), intraductal pancreatic mucinous neoplasm (n = 1), serous cystadenoma (n = 1), and NET (n = 1). Of the 55 patients with PDAC, MARS1 staining was positive in 54 and negative in 1 (Figure 3).

Among the 39 patients with specimens designated as atypia by CCM, the final clinicopathologic diagnoses were PDAC (n = 36), mucinous cystic neoplasm (n = 1), solid pseudopapillary neoplasm (n = 1), and mixed acinar-neuroendocrine carcinoma (n = 1). Of the 36 patients with PDAC, MARS1 staining was positive in 35 and negative in 1 (Figure 4).

The specimen was designated as negative by CCM in one patient, in whom the final clinicopathologic diagnosis was autoimmune pancreatitis and MARS1 staining was negative.

MARS1 staining allowed identification of malignancy by detecting high MARS1 expression in samples that were indeterminate (atypia or suspicious for malignancy) according to CCM (Table 2, Figure 4). The sensitivity, specificity, PPV, NPV, and accuracy for detecting malignancy were 97.4%, 50%, 97.4%, 50.0%, and 95.0%, respectively, for MARS1 IF staining and 79.1%, 37.5%, 96.0%, 8.6%, and 77.0%, respectively, for CCM (Table 3).

MARS1 IF significantly outperformed CCM in terms of sensitivity (*p* < 0.0001), accuracy (*p* < 0.0001), and NPV (*p* = 0.166).

## 4. Discussion

For detecting pancreatic cancer, EUS-FNA cytology has a pooled sensitivity of approximately 77% (66–86%) and pooled specificity of approximately 98% (78–100%) [19]. Several technical improvements have been developed to increase the sensitivity of EUS-FNA, including fanning, suction, stylet use, and rapid on-site evaluation (ROSE). EUS imaging enhancements, such as contrast-enhanced harmonic and elastography, may increase the diagnostic yield of EUS-FNA by targeting the needle away from necrotic areas [20,21,22,23,24]. In clinical practice, 22- and 25-gauge needles are the most commonly used for EUS-guided fine-needle aspiration (EUS-FNA), and current evidence suggests no significant superiority of the 25-gauge needle over the 22-gauge needle in terms of diagnostic performance for pancreatic masses [25]. A meta-analysis examining repeat EUS-FNA after an initial nondiagnostic procedure found a sensitivity and specificity of 83% (64–93%) and 98% (80–100%), respectively, when ROSE was used and 65% (57–73%) and 94% (31–100%), respectively, when it was unavailable [19].

Compared to FNA needles, FNB needles demonstrate superior performance in diagnosis of solid pancreatic masses, offering higher diagnostic accuracy, improved sample adequacy, greater DNA yield, and diagnostic efficiency with fewer passes, without increasing adverse events [26,27]. Sensitivity has continued to improve with the development of various FNB needle designs. The pooled sensitivity and specificity of FNB are 84% (95% CI, 0.82–0.87) and 98% (95% CI, 0.93–1.00), respectively [28]. Particularly, the Franseen needle demonstrates a sensitivity of 93.9% (95% CI, 90.8–98.4%), while the Fork-tip needle shows a sensitivity of 90.4% (95% CI, 81.6–98%). Notably, the specificity of these needles is reported to be close to 100% [29,30,31]. Based on these benefits and moderate-quality evidence, their use is strongly recommended [26].

Despite attempts to improve tumor sampling methodology, morphologic criteria cannot reliably discriminate neoplastic from inflammatory or reactive cellular changes. Therefore, a molecular assay is necessary to improve the diagnostic performance of EUS-FNA. Various markers are being studied for immunohistochemical staining of pancreatic masses or cysts [32], and to distinguish pancreatic cancer from chronic pancreatitis.

Mesothelin, S100 calcium-binding protein P (S100P), insulin-like growth factor II mRNA binding protein3 (IMP3), and mucin are cancer-specific markers expressed in pancreatic cancer. Immunohistochemical staining of cytology samples for S100P had a sensitivity and specificity of 96.4% and 93.3%, respectively, for diagnosing PADC, whereas staining for IMP3 had a sensitivity and specificity of 91.2% and 86.7%, respectively [33]. Together, the two markers had a diagnostic accuracy of 89%. When the amount of S100P (quantified by enzyme-linked immunosorbent assay) was combined with pathology findings, the sensitivity was 94.4% and specificity was 88.9% [34]. However, the study had a small sample size, and the amount of pancreatic tissue collected per specimen was limited, so the sensitivity of pathology findings alone was low (77.8%).

When simultaneous immunohistochemical staining for IMP3 and mesothelin was performed on pancreatic cancer, chronic pancreatitis, and normal pancreatic tissues, the sensitivity was 85% and specificity was 90% [35]. A meta-analysis found that mesothelin staining alone had a sensitivity of 71% and specificity of 88% for diagnosing pancreatic cancer [36]. Although IMP3 appears to have a higher sensitivity and specificity than mesothelin based on the available literature, no firm conclusions can be made. In this study, the sensitivity of MARS1 was 97.4%, comparable to that of S100P; however, its specificity was relatively low at 50%. Currently, an immunocytochemistry staining method for MARS1 is under development. It is anticipated that the diagnostic performance may be further improved in the future through combined staining with previously established markers such as S100P.

High MARS1 expression in carcinoma can be used to diagnose cancer in indeterminate specimens [5,16,17]. We previously reported that the high sensitivity and accuracy of MARS1 IF staining in brush cytology specimens enabled detection of biliary malignancy in indeterminate biliary strictures [17]. In the present study, MARS1 IF staining was more sensitive than conventional Pap staining (97.4 vs. 73.2%) in diagnosing pancreatic cancer. Dual MARS1/CD45 IF staining has good diagnostic performance and may be a valuable addition to cytology tests in determining lymph node metastasis of non-small-cell lung cancer (NSCLC). Combining MARS1 staining with conventional cytology also improves the accuracy of diagnosis in patients with suspected lung cancer [16]. Dual IF staining using combinations of MARS1, ARS-interacting multifunctional protein-lacking exon 2 (AIMP2-DX2), and/or pan-cytokeratin is also an effective diagnostic tool, which can improve lung cancer diagnostic yield by complementing conventional cytology [5].

We previously investigated the clinical significance of MARS1 expression in PDAC [14]. We found that MARS1 expression was clearly higher in pancreatic cancer than in normal pancreatic ductal tissues. Furthermore, patients with high MARS1 expression had shorter overall and recurrence-free survivals than those with low expression [14]. MARS1 is frequently overexpressed in NSCLC. This overexpression is correlated with increased mammalian target of rapamycin complex1 activity and is associated with poorer clinical outcomes, suggesting that MARS1 may be a potential therapeutic target [11]. Autoantibodies targeting both AIMP2-DX2 and AIMP2 have been detected in human serum, with an elevated AIMP2-DX2 to AIMP2 ratio correlating with poorer clinical outcomes in lung cancer patients. Similarly, elevated levels of MARS1 expression in breast cancer tissue have been linked to reduced survival, indicating its potential as both a diagnostic and prognostic biomarker in that context [37].

In our study, we observed pronounced MARS1 expression in pancreatic ductal adenocarcinoma (PDAC), suggesting its utility as a diagnostic indicator. Immunohistochemical analysis revealed strong MARS1 staining in resected PDAC tissues, contrasting sharply with the minimal expression in non-cancerous samples [14]. Further investigation is warranted to clarify whether MARS1 contributes to the initiation or advancement of PDAC. Notably, we extended the application of our previously developed MARS1 immunofluorescence protocol—originally used for brush cytology samples from biliary strictures—to EUS-FNA cytology samples of pancreatic masses. The sensitivity of MARS1 IF staining for detecting malignancy in patients with bile duct strictures was 98.1% in our initial single-center study and 93.6% in our subsequent multicenter study, which were superior to the sensitivity values for CCM in these studies (70.4% and 73.2%, respectively) [17]. Our current results likewise showed that the sensitivity was significantly higher with MARS1 IF staining than with conventional Pap staining (97.4% vs. 79.1%) in pancreatic EUS-FNA specimens. This superior performance reflects the ability to differentiate malignant and benign conditions in samples designated as atypia on Pap staining. Using MARS1 IF staining to accurately characterize pancreatic masses designated as indeterminate on conventional analysis of EUS-FNA samples may reduce unnecessary surgical interventions and guide therapeutic strategies. Of the 151 patients with a final diagnosis of PDAC, 36 (23.8%) had atypia on Pap staining, and of these 36 patients, 35 (97.2%) were MARS1 IF–positive. Thus, the higher sensitivity of MARS1 IF staining appears to reflect its ability to confirm malignancy in patients with PDAC who have indeterminate CCM results.

In this study, the sensitivity and specificity of FNA cytology with MARS1 staining were 97.4% and 50%, respectively. While the sensitivity is comparable to that of FNB, the specificity is notably lower. In contrast, FNA cytology with Pap staining showed a sensitivity of 79.1% and a specificity of 37.5%. Although FNA cytology with MARS1 staining demonstrates similar sensitivity to FNB, its specificity remains lower. Nevertheless, it shows higher sensitivity and specificity compared to FNA cytology with Pap staining. Therefore, FNA cytology with MARS1 staining may serve as a useful diagnostic alternative for pancreatic masses when FNB is not feasible.

This study has limitations, including the relatively small number of cytology specimens, particularly negative specimens. Among the 161 included patients, only 10 did not have PDAC, 6 of whom had specimens negative for MARS1. The sensitivity of EUS-FNA was high, as most patients had PDAC. Furthermore, although we enrolled 198 patients, 33 with nondiagnostic or insufficient diagnosis results on CCM were subsequently excluded. Excluding these patients may have increased the sensitivity or accuracy of cytologic diagnosis for malignancy, it should not have affected the MARS1 expression results. To confirm our results, a large-scale prospective study including a sufficient number of patients with a benign pancreatic mass is necessary. Another limitation was that separate slide preparations were required for Pap staining and MARS1 IF, even though both were processed using the same ThinPrep system. Because these staining procedures could not be conducted concurrently on a single slide, there is a possibility that the cellular clusters analyzed differed between the two methods. This discrepancy may have influenced the comparative evaluation of paired specimens. Simultaneously comparing multiple slides obtained from a single EUS-FNA specimen using the Surepath system would help overcome this limitation.

## 5. Conclusions

The accurate diagnosis of pancreatic masses is essential for optimal treatment and prognosis. Our new IF-staining method using a MARS1-specific antibody improved the diagnostic performance in EUS-FNA specimens from patients with a pancreatic mass. Our results demonstrated that MARS1 may serve as a novel diagnostic marker for PDAC. Further prospective studies are necessary to validate our findings.

## Figures and Tables

**Figure 1 diagnostics-15-01783-f001:**
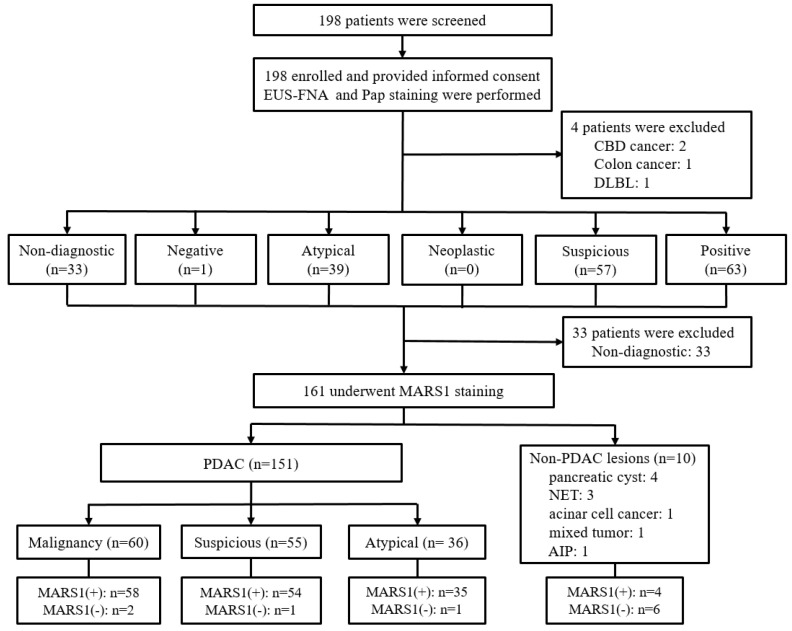
Patient flowchart. AIP, autoimmune pancreatitis; CBD, common bile duct; DLBL, diffuse large B-cell lymphoma; EUS-FNA, endoscopic ultrasound-guided fine-needle aspiration; MARS1, methionyl-tRNA synthetase 1; NET, neuroendocrine tumor; Pap, Papanicolaou; PDAC, pancreatic ductal adenocarcinoma.

**Figure 2 diagnostics-15-01783-f002:**
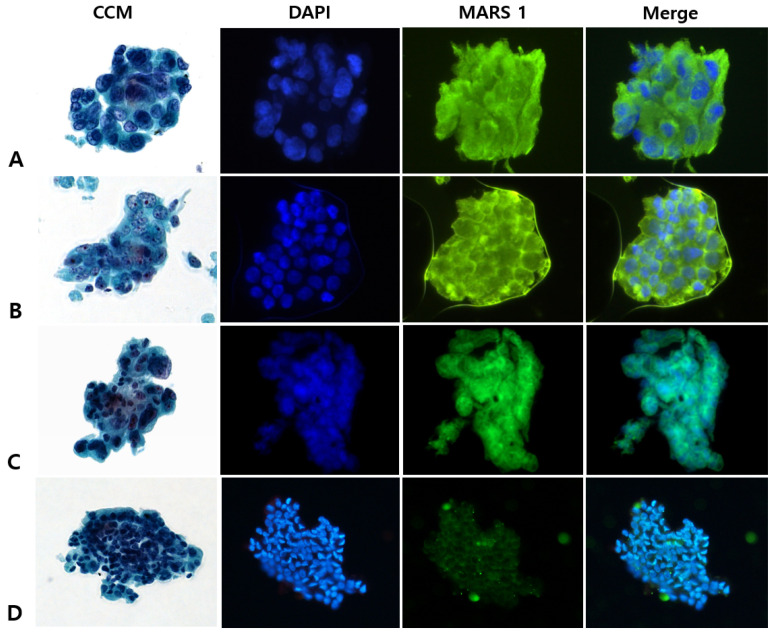
Representative photographs of conventional cytologic staining (CCM) and immunofluorescence (IF) staining for methionyl-tRNA synthetase 1 (MARS1) in pancreatic ductal adenocarcinoma specimens that were positive for malignancy by CCM (final diagnosis, malignancy). (**A**–**C**), Examples of the cytology specimens from 58 patients that showed strong and diffuse MARS1 signals in the cytoplasm (400×). (**D**), Examples of the cytology specimens from 2 patients that were negative or weakly positive for MARS1 (400×). CCM represents Papanicolaou staining, DAPI represents nuclear staining with 4′,6-diamidino-2-phenylindole, MARS1 represents MARS1 IF staining, and Merge represents the combined MARS1 and DAPI staining images.

**Figure 3 diagnostics-15-01783-f003:**
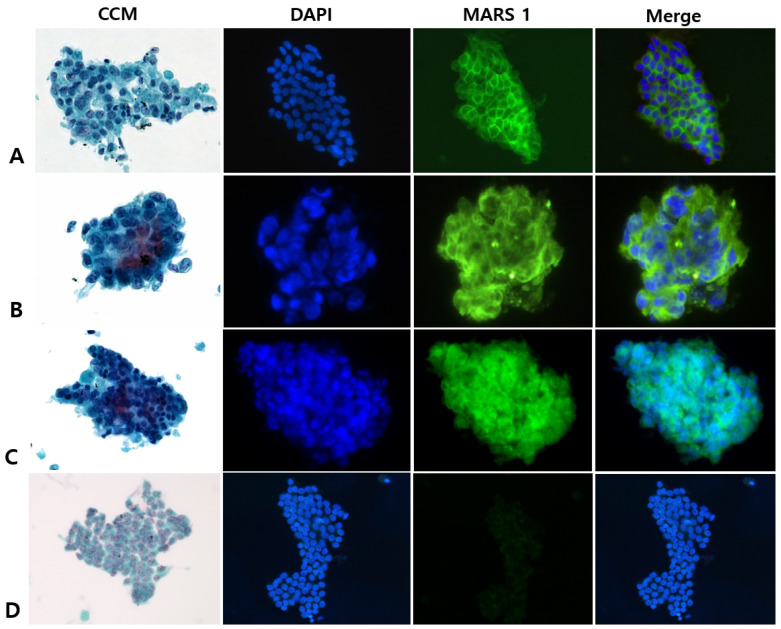
Representative photographs of conventional cytologic staining (CCM) and immunofluorescence (IF) staining for methionyl-tRNA synthetase 1 (MARS1) in pancreatic ductal adenocarcinoma specimens that were designated as suspicious for malignancy by CCM (final diagnosis, malignancy). (**A**–**C**), Examples of the cytology specimens from 54 patients that showed strong and diffuse MARS1 signals in the cytoplasm (400×). (**D**), Photographs of the cytology specimens from 1 patient that were negative or weakly positive for MARS1 (400×). CCM represents Papanicolaou staining, DAPI represents nuclear staining with 4′,6-diamidino-2-phenylindole, MARS1 represents MARS1 IF staining, and Merge represents the combined MARS1 and DAPI staining images.

**Figure 4 diagnostics-15-01783-f004:**
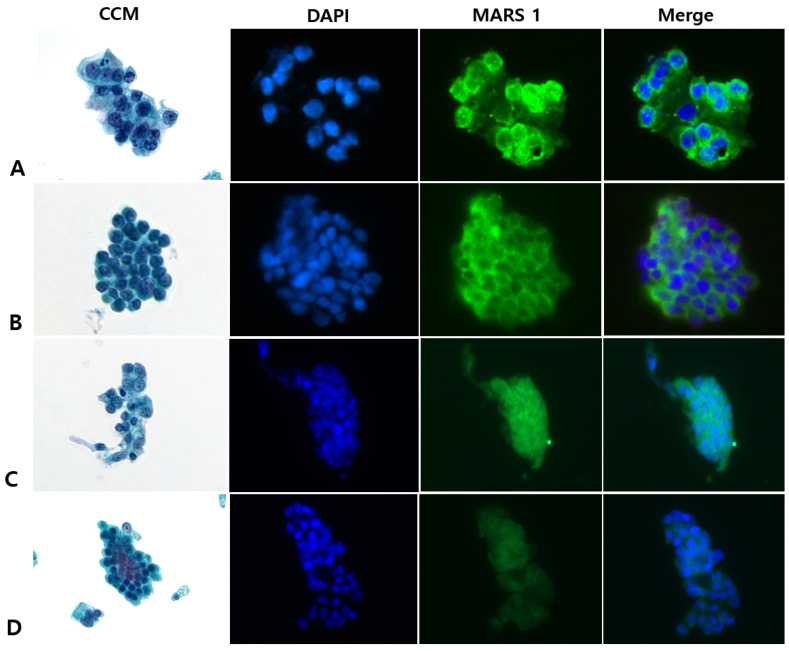
Representative photographs of conventional cytological staining (CCM) and immunofluorescence (IF) staining for methionyl-tRNA synthetase 1 (MARS1) in pancreatic ductal adenocarcinoma specimens that were designated as atypia by CCM (final diagnosis, malignancy). (**A**–**C**), Examples of the cytology specimens from 35 patients that showed strong and diffuse MARS1 signals in the cytoplasm (400×). (**D**), Photographs of the cytology specimens from 1 patient with a negative or weakly positive signal for MARS1 (400×). CCM represents Papanicolaou staining, DAPI represents nuclear staining with 4′,6-diamidino-2-phenylindole, MARS1 represents MARS1 IF staining, and Merge represents the combined MARS1 and DAPI staining images.

**Table 1 diagnostics-15-01783-t001:** Patient characteristics.

Characteristic	N (%)
**Total number of patients**	161
**Age**, y (mean ± SD)	65.6 ± 10.3
**Sex**, n (M:F)	82:79
**Diagnosis**, n (%)	
** Pancreatic ductal adenocarcinoma**	151 (93.8)
** Pancreatic cyst**	4 (2.5)
Intraductal pancreatic muninous neoplasm	1
Mucinous cystic neoplasm	1
Serous cystadenoma	1
Solid papillary neoplasm	1
** Other pancreatic lesions**	6 (3.7)
Pancreatic neuroendocrine tumor	3
Mixed acinar-neuroendocrine carcinoma	1
Acinar cell carcinoma	1
Autoimmune pancreatitis	1

Data are number, number (percentage), or mean (standard deviation, SD). MARS1, methionyl-tRNA synthetase 1; Pap, Papanicolaou.

**Table 2 diagnostics-15-01783-t002:** Correlation of cytological assessment and methionyl-tRNA synthetase 1 (MARS1) with the final clinicopathological diagnosis.

Diagnosis	Pap Staining †	MARS1 Immunostaining *	
		Positive	Negative
**PDAC** (n = 151)	malignancy (n = 60)	58	2
	suspicious (n = 55)	54	1
	atypical (n = 36)	35	1
**Pancreatic cyst** (n = 4)			
IPMN (n = 1)	suspicious (n = 1)	1	0
MCN (n = 1)	atypical (n = 1)	0	1
SCA (n = 1)	suspicious (n = 1)	0	1
SPN (n = 1)	atypical (n = 1)	1	0
**Other pancreatic lesions** (n = 6)			
NET (n = 3)	malignancy (n = 2)	0	2
	suspicious (n = 1)	1	0
Acinar cell carcinoma (n = 1)	malignancy (n = 1)	0	1
Mixed AC-NEC (n = 1) ‡	atypical (n = 1)	1	0
AIP (n = 1)	negative (n = 1)	0	1

AIP, autoimmune pancreatitis; IF, immunofluorescence; IPMN, intraductal papillary mucinous neoplasm; MARS1, methionyl-tRNA synthetase 1; MCN, mucinous cystic neoplasm; Mixed AC-NEC, mixed acinar-neuroendocrine carcinoma; NET, neuroendocrine tumor; PDAC, pancreatic ductal adenocarcinoma; SCA, serous cystadenoma; SPN, solid pseudopapillary neoplasm. * *p* < 0.0001 vs. Pap staining, Cochran–Mantel–Haenszel test. † Results of Papanicolaou staining (a conventional cytologic staining method). ‡ Mixed AC-NEC containing acinar cell carcinoma (40%) and neuroendocrine tumor (60%) cells.

**Table 3 diagnostics-15-01783-t003:** Diagnostic performance of the conventional cytological staining method (CCM) and methionyl-tRNA synthetase 1 (MARS1) immunofluorescence staining.

	Sensitivity	Specificity	PPV	NPV	Accuracy
Conventional cytology method	79.1 (72.6–85.5)	37.5 (39.5–71.1)	96.0 (92.6–99.0)	8.6 (0–17.8)	77.0 (70.5–83.5)
MARS1 immunostaining	97.4 (94.8–99.9)	50 (15.4–84.6)	97.4 (94.9–99.9)	50 (15.4–84.6)	95.0 (91.7–98.4)
*p* value *	<0.0001	0.6506	0.4030	0.0166	<0.0001

CCM, conventional cytologic staining method; MARS1, methionyl-tRNA synthetase 1; NPV, negative predictive value; PPV, positive predictive value. Values expressed as % (95% confidence interval) * *p* value for CCM vs. MARS1.

## Data Availability

The data used in this study are available in this article.

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
