# Peer review of "A New Staining Method Using Methionyl-tRNA Synthetase 1 Antibody for Endoscopic Ultrasound-Guided Fine-Needle Aspiration Cytology of Pancreatic Cancer"

_diagnostics, 2025, doi:10.3390/diagnostics15141783_

Round 1

Reviewer 1 Report

Comments and Suggestions for Authors

The study is interesting and timely. I have some concerns:

1) The authors should comment in the discussion that the current practice is to use FNB over FNA, particularly when ROSE is not available.

2) Could the authors be sure that IHC analysis with FNA samples is as well effective as with FNB samples?

3) THe authors should comment that 22 and 25G FNA needles are the most used in the clinical practice and they are equally effective (in this regard cite the meta-analysis: PMID: 29026598)

Author Response

â–£ For Reviewer 1

We appreciate your thoughtful comments regarding our manuscript.

â–£ Evaluations

Comments by the reviewers:

Reviewer 1

The study is interesting and timely. I have some concerns:

1) The authors should comment in the discussion that the current practice is to use FNB over FNA, particularly when ROSE is not available.

Answer) As the reviewer mentioned, the recent introduction of FNB has increased the diagnostic rate for pancreatic masses compared to FNA. We added the following to the discussion about these aspects:

‘Compared to FNA needles, FNB needles demonstrate superior performance in di-agnosis of solid pancreatic masses, offering higher diagnostic accuracy, improved sample adequacy, greater DNA yield, and diagnostic efficiency with fewer passes, without increasing adverse events [28,29]. Sensitivity has continued to improve with the development of various FNB needle designs. The pooled sensitivity and specificity of FNB are 84% (95% CI, 0.82–0.87) and 98% (95% CI, 0.93–1.00), respectively [30]. Particularly, the Franseen needle demonstrates a sensitivity of 93.9% (95% CI, 90.8%–98.4%), while the Fork-tip needle shows a sensitivity of 90.4% (95% CI, 81.6%–98%). Notably, the specificity of these needles is reported to be close to 100% [31-33]. Based on these benefits and moderate-quality evidence, their use is strongly recommended [28].’

Reference

  1. Committee, A.S.o.P.; Machicado, J.D.; Sheth, S.G.; Chalhoub, J.M.; Forbes, N.; Desai, M.; Ngamruengphong, S.; Papachristou, G.I.; Sahai, V.; Nassour, I. et al. American society for gastrointestinal endoscopy guideline on the role of endoscopy in the diagnosis and management of solid pancreatic masses: Summary and recommendations. Gastrointestinal endoscopy 2024, 100, 786–796.
  2. Chong, C.C.; Pittayanon, R.; Pausawasdi, N.; Bhatia, V.; Okuno, N.; Tang, R.S.; Cheng, T.Y.; Kuo, Y.T.; Oh, D.; Song, T.J. et al. Consensus statements on endoscopic ultrasound-guided tissue acquisition. Guidelines from the asian endoscopic ultrasound group. Digestive endoscopy : official journal of the Japan Gastroenterological Endoscopy Society 2024, 36, 871–883.
  3. Yang, Y.; Li, L.; Qu, C.; Liang, S.; Zeng, B.; Luo, Z. Endoscopic ultrasound-guided fine needle core biopsy for the diagnosis of pancreatic malignant lesions: A systematic review and meta-analysis. Sci Rep 2016, 6, 22978.
  4. Gkolfakis, P.; Crino, S.F.; Tziatzios, G.; Ramai, D.; Papaefthymiou, A.; Papanikolaou, I.S.; Triantafyllou, K.; Arvanitakis, M.; Lisotti, A.; Fusaroli, P. et al. Comparative diagnostic performance of end-cutting fine-needle biopsy needles for eus tissue sampling of solid pancreatic masses: A network meta-analysis. Gastrointestinal endoscopy 2022, 95, 1067–1077 e1015.
  5. van Riet, P.A.; Erler, N.S.; Bruno, M.J.; Cahen, D.L. Comparison of fine-needle aspiration and fine-needle biopsy devices for endoscopic ultrasound-guided sampling of solid lesions: A systemic review and meta-analysis. Endoscopy 2021, 53, 411–423.
  6. Levine, I.; Trindade, A.J. Endoscopic ultrasound fine needle aspiration vs fine needle biopsy for pancreatic masses, subepithelial lesions, and lymph nodes. World journal of gastroenterology 2021, 27, 4194–4207.

2) Could the authors be sure that IHC analysis with FNA samples is as well effective as with FNB samples?

Answer) The pooled sensitivity and specificity of FNB reported to date are 84% (95% CI, 0.82–0.87) and 98% (95% CI, 0.93–1.00), respectively [4]. Sensitivity has improved with the development of various FNB needle designs. Specifically, the Franseen needle shows a sensitivity of 93.9% (95% CI, 90.8%–98.4%), and the Fork-tip needle shows a sensitivity of 90.4% (95% CI, 81.6%–98%). Notably, the specificity of these needles approaches 100% [3,5,6].

In this study, the sensitivity and specificity of MARS1 staining in FNA cytology were 97.4% and 50%, respectively. While the sensitivity is comparable to that of FNB, the specificity is lower. In contrast, Pap staining in FNA cytology showed a sensitivity of 79.1% and a specificity of 37.5%.

FNA cytology with MARS1 staining demonstrates similar sensitivity to FNB and superior performance compared to FNA with Pap staining, particularly in terms of both sensitivity and specificity. Therefore, FNA cytology with MARS1 staining may be a useful diagnostic option for pancreatic masses when FNB is not feasible.

This is described as follows.

‘In this study, the sensitivity and specificity of FNA cytology with MARS1 staining were 97.4% and 50%, respectively. While the sensitivity is comparable to that of FNB, the specificity is notably lower. In contrast, FNA cytology with Pap staining showed a sensitivity of 79.1% and a specificity of 37.5%. Although FNA cytology with MARS1 staining demonstrates similar sensitivity to FNB, its specificity remains lower. Nevertheless, it shows higher sensitivity and specificity compared to FNA cytology with Pap staining. Therefore, FNA cytology with MARS1 staining may serve as a useful diagnostic alternative for pancreatic masses when FNB is not feasible.’

3) The authors should comment that 22 and 25G FNA needles are the most used in the clinical practice and they are equally effective (in this regard cite the meta-analysis: PMID: 29026598)

Answer) As the reviewer pointed out, 22G and 25G FNA needles are the most used in the clinical practice and they are equally effective. We wrote about these parts as follows.

‘In clinical practice, 22- and 25-gauge needles are the most commonly used for EUS-guided fine-needle aspiration (EUS-FNA), and current evidence suggests no sig-nificant superiority of the 25-gauge needle over the 22-gauge needle in terms of diag-nostic performance for pancreatic masses [27].’

Reference

  1. Facciorusso, A.; Stasi, E.; Di Maso, M.; Serviddio, G.; Ali Hussein, M.S.; Muscatiello, N. Endoscopic ultrasound-guided fine needle aspiration of pancreatic lesions with 22 versus 25 gauge needles: A meta-analysis. United European Gastroenterol J 2017, 5, 846–853.

Reviewer 2 Report

Comments and Suggestions for Authors

Overall, this study does a good job at introducing a novel IHC marker, MARS1, for diagnosing pancreatic cancer (PDAC) from EUS-FNA samples. MARS1 as a marker has a high sensitivity and been previously shown to correlate with poor disease progression and outcomes may become a useful addition to the oncology armamentarium. In general, there are a few spots where numbers are not clear or inconsistent and should be verified prior to publication. Additionally, the discussion has a point that should be supported with additional data or increased discussion to increase the clinical relevance of the current study within prior literature.

  1. Line 96, you state that light microscopy was used to look at the IHC stained samples. However, normally light microscopy is used for non-fluorescent analysis, and IHC uses fluorescent microscopy. Please verify.
  2. Lines 113-114 state there were “6 slides from 3 sample tubes. Three slides from each tube...” Then, with 3 slides from each sample tube there should be 9 slides. Please verify the correct sample numbers.
  3. Discussion paragraph 298-313 feels more like an introduction to the physiology of MARS1 than a discussion on the results of this study. I would suggest either shortening this in this section or translating it to the introduction section.
  4. Line 318 states that the sensitivity for the MARS1 IF is 93.6% in the present study, while lines 42, 257, 350 and Table 3 its 97.4%. Line 346 has the 93.6% sensitivity tied to a prior study, which I assume is the same for line 71. Please clarify and verify correct values.
  5. Discussion paragraph from lines 282-297 focuses on the sensitivity and specificity of other IHC tested markers for PDAC. Given that the focus of the conclusion and impact of this study is in utilizing MARS1 IHC for diagnostics, there is a connection missing between the existing knowledge and the present study. It appears that the present study has similar sensitivities, with notably lower specificity. It would improve the context of the study to include IHC of their samples with one of these noted markers, perhaps the S100P given its higher reported sensitivity and specificity. If infeasible, there should be a more direct comparison of these values in the discussion.

Author Response

â–£ For Reviewer 2

We appreciate your thoughtful comments regarding our manuscript.

â–£ Evaluations

Comments by the reviewers:

Reviewer 2

Overall, this study does a good job at introducing a novel IHC marker, MARS1, for diagnosing pancreatic cancer (PDAC) from EUS-FNA samples. MARS1 as a marker has a high sensitivity and been previously shown to correlate with poor disease progression and outcomes may become a useful addition to the oncology armamentarium. In general, there are a few spots where numbers are not clear or inconsistent and should be verified prior to publication. Additionally, the discussion has a point that should be supported with additional data or increased discussion to increase the clinical relevance of the current study within prior literature.

  1. Line 96, you state that light microscopy was used to look at the IHC stained samples. However, normally light microscopy is used for non-fluorescent analysis, and IHC uses fluorescent microscopy. Please verify.

Answer) As the reviewer pointed out, light microscopy cannot be used for fluorescence analysis. Light microscopy was used in surgical specimens. MARS1 staining can be done by both IHC and fluorescent staining. IHC staining was performed in surgical specimens, and cytology was done by IF staining. This is described in method section below.

‘MARS1 staining used IHC and fluorescent staining. IHC staining was performed on surgical specimens, and cytology was performed using IF staining.’

  1. Lines 113-114 state there were “6 slides from 3 sample tubes. Three slides from each tube...” Then, with 3 slides from each sample tube there should be 9 slides. Please verify the correct sample numbers.

Answer) Two slides were produced from one tube, for a total of six slides. One slide from each tube was collected, and Pap staining was performed on three slides, and MARS1 staining was performed on the remaining three slides. As the reviewer pointed out, there may have been confusion, so it has been revised as follows.

‘Two slides were produced from one tube, for a total of six slides. One slide was collected from each tube, and Pap staining was performed on three slides, and MARS1 staining was performed on the remaining three slides.’

  1. Discussion paragraph 298-313 feels more like an introduction to the physiology of MARS1 than a discussion on the results of this study. I would suggest either shortening this in this section or translating it to the introduction section.

Answer) As the reviewer pointed out, we think this content would be suitable for introduction. I shortened the content as follows and moved it to the introduction section.

‘MARS1 is highly expressed in multiple cancer types and is increasingly recog-nized for its role in tumorigenesis and cancer progression [7-15]. In pancreatic ductal adenocarcinoma (PDAC), elevated MARS1 expression correlates with poor prognosis, and in our previous study, it was identified—alongside lymph node metastasis—as an independent predictor of unfavorable outcomes in PDAC patients [16]. Mechanisti-cally, MARS1 contributes to oncogenesis through various pathways: it enhances en-zymatic activity in colon cancer cells [17], and overexpression of its substrate, initiator tRNAiMet, is sufficient to induce malignant transformation [18]. Furthermore, MARS1 stabilizes cyclin-dependent kinase 4 (CDK4) by forming a complex with heat shock protein 90, protecting CDK4 from proteasomal degradation and promoting cell cycle progression [5]. Conversely, suppression of MARS1 leads to CDK4 depletion and cell cycle arrest at the G0/G1 phase. MARS1 also competes with the tumor suppressor p16^INK4a for binding to the N-terminal domain of CDK4, further highlighting its potential oncogenic role [5].’

  1. Line 318 states that the sensitivity for the MARS1 IF is 93.6% in the present study, while lines 42, 257, 350 and Table 3 its 97.4%. Line 346 has the 93.6% sensitivity tied to a prior study, which I assume is the same for line 71. Please clarify and verify correct values.

Answer) In this study, the sensitivity for the MARS1 IF is 97.4%. The typo was corrected to 97.4% as follows.

‘In the present study, MARS1 IF staining was more sensitive than conventional Pap staining (97.4 vs. 79.1%) in diagnosing pancreatic cancer.’

  1. Discussion paragraph from lines 282-297 focuses on the sensitivity and specificity of other IHC tested markers for PDAC. Given that the focus of the conclusion and impact of this study is in utilizing MARS1 IHC for diagnostics, there is a connection missing between the existing knowledge and the present study. It appears that the present study has similar sensitivities, with notably lower specificity. It would improve the context of the study to include IHC of their samples with one of these noted markers, perhaps the S100P given its higher reported sensitivity and specificity. If infeasible, there should be a more direct comparison of these values in the discussion.

Answer) Immunohistochemical staining of cytology samples for S100P demonstrated a sensitivity and specificity of 96.4% and 93.3%, respectively, for diagnosing PDAC. When the quantitative level of S100P (measured by enzyme-linked immunosorbent assay) was combined with pathological findings, the sensitivity and specificity were 94.4% and 88.9%, respectively. However, due to the small sample size and limited amount of pancreatic tissue obtained per specimen, the sensitivity of pathological findings alone was relatively low (77.8%). As no further studies have been conducted, direct comparison with MARS1 remains challenging.

MARS1 is currently applicable for both immunofluorescence (IF) and immunohistochemical (IHC) staining. For practical clinical use, we are implementing IHC staining in cytology and conducting a comparative study between IF and immunocytochemistry (ICC) staining. Although a direct comparison with S100P is difficult at this stage, we are developing an ICC staining method for MARS1 and have incorporated relevant findings into our ongoing research.

We wrote about these parts as follows.

‘In this study, the sensitivity of MARS1 was 97.4%, comparable to that of S100P; however, its specificity was relatively low at 50%. Currently, an immunocytochemistry staining method for MARS1 is under development. It is anticipated that the diagnostic performance may be further improved in the future through combined staining with previously established markers such as S100P.’

Round 2

Reviewer 1 Report

Comments and Suggestions for Authors

The manuscript is fine now. Thank you!

Reviewer 2 Report

Comments and Suggestions for Authors

Thank you for making the changes. I believe the manuscript is sufficiently improved and have no further comments.